ⓐ | Open Peer Review | Environmental Microbiology | Research Article

# Metagenomic assembled genomes indicated the potential application of hypersaline microbiome for plant growth promotion and stress alleviation in salinized soils

Kiran Dindhoria,[1,2] Raghawendra Kumar,[1] Bhavya Bhargava,[1] Rakshak Kumar[1,2]

**ABSTRACT** Climate change is causing unpredictable seasonal variations globally. Due to the continuously increasing earth's surface temperature, the rate of water evaporation is enhanced, conceiving a problem of soil salinization, especially in arid and semi-arid regions. The accumulation of salt degrades soil quality, impairs plant growth, and reduces agricultural yields. Salt-tolerant, plant-growth-promoting microorganisms may offer a solution, enhancing crop productivity and soil fertility in salinized areas. In the current study, genome-resolved metagenomic analysis has been performed to investigate the salt-tolerating and plant growth-promoting potential of two hypersaline ecosystems, Sambhar Lake and Drang Mine. The samples were co-assembled independently by Megahit, MetaSpades, and IDBA-UD tools. A total of 67 metagenomic assembled genomes (MAGs) were reconstructed following the binning process, including 15 from Megahit, 26 from MetaSpades, and 26 from IDBA_UD assembly tools. As compared to other assemblers, the MAGs obtained by MetaSpades were of superior quality, with a completeness range of 12.95%–96.56% and a contamination range of 0%–8.65%. The medium and high-quality MAGs from MetaSpades, upon functional annotation, revealed properties such as salt tolerance (91.3%), heavy metal tolerance (95.6%), exopolysaccharide (95.6%), and antioxidant (60.86%) biosynthesis. Several plant growth-promoting attributes, including phosphate solubilization and indole-3-acetic acid (IAA) production, were consistently identified across all obtained MAGs. Conversely, characteristics such as iron acquisition and potassium solubilization were observed in a substantial majority, specifically 91.3%, of the MAGs. The present study indicates that hypersaline microflora can be used as bio-fertilizing agents for agricultural practices in salinized areas by alleviating prevalent stresses.

**IMPORTANCE** The strategic implementation of metagenomic assembled genomes (MAGs) in exploring the properties and harnessing microorganisms from ecosystems like hypersaline niches has transformative potential in agriculture. This approach promises to redefine our comprehension of microbial diversity and its ecosystem roles. Recovery and decoding of MAGs unlock genetic resources, enabling the development of new solutions for agricultural challenges. Enhanced understanding of these microbial communities can lead to more efficient nutrient cycling, pest control, and soil health maintenance. Consequently, traditional agricultural practices can be improved, resulting in increased yields, reduced environmental impacts, and heightened sustainability. MAGs offer a promising avenue for sustainable agriculture, bridging the gap between cutting-edge genomics and practical field applications.

**KEYWORDS** hypersaline ecosystems, metagenomic assembled genomes, salt stress, heavy metal stress, plant growth promotion, salinized soil

Address correspondence to Rakshak Kumar, rakshak@ihbt.res.in.

The authors of declare no conflict of interest.

Hypersaline environments are frequently referred to as "poly-extreme ecosystems" because of their exposure to multiple stress conditions such as high salinities, high osmotic pressure, low water availability, and environmental variables such as temperature, pH, and dissolved oxygen (1). They pose several challenges to life due to their harsh conditions. Despite their hostile conditions, they are capable of supporting a surprising diversity of specialized organisms, including halotolerant and halophilic microorganisms. All three domains of life, Archaea, Bacteria, and Eukarya, have halophilic and extremely halotolerant members that live in hypersaline ecosystems containing salt concentrations at or above the NaCl saturation point. These halophilic or extremely salt-tolerant microorganisms employ various methods to resist high salt concentrations and, in many cases, modify their physiology in response to variations in salt concentrations in their surroundings. To maintain the osmotic pressure, halophiles generally use "salt in strategy," in which inorganic ions $Cl^-$ and $K^+$ ions are accumulated inside the cell (2). In these conditions, the proteome of the microorganisms requires a high internal salt content for efficient folding and operation of both enzymes and proteins. The intake or synthesis of osmolytes is another method used by hypersaline bacteria. It is sometimes referred to as the compatible solute or low-salt-in strategy. Osmolytes/compatible solutes are small chemical molecules that do not impact cellular metabolism (3). Sugars, polyols, amino acids, and their derivatives are the most common osmolytes. Microorganisms synthesize compatible solutes such as glycine, betaine, ectoine, glycerol, trehalose, sucrose, etc., when the environment contains high salt content (4, 5). They are strong water structure formers that are probably excluded from proteins' hydration shells, lowering water activity coefficients and stabilizing the hydration shell (6). In addition to assisting the hypersaline microflora in maintaining regular metabolic processes and osmotic balance maintenance, the compatible solutes also prevent microbial proteins from denaturation, increasing their capacity to withstand significant changes in the surrounding saline environment (7). Since hypersaline microflora are well adapted to extreme salt conditions, they have garnered significant interest in various biotechnological applications (8–11). Their diverse metabolic capabilities and resilience to extreme environments have made them valuable candidates. There are several reports exploring the microbial and metabolic diversity of various hypersaline ecosystems. In 2021, Bueno de Mesquita and colleagues uncovered a new genus, *Methanosalis*, during their investigation of the salt pond in San Francisco (12). In addition, the research group of Chakraborty and Kurth delved into the xenobiotic-degrading and carbon fixation capabilities of Lonar Lake in India and La Brava and Tebenquiche lakes in Salar de Atacama, Chile, respectively (13, 14). Collectively, these studies contribute to our understanding of the unique microbial communities and metabolic processes present in hypersaline environments across different geographical locations. The hypersaline microbes also break down complex organic matter, releasing nutrients like nitrogen and phosphorus that are needed for plant growth. The biogeochemical functions of hypersaline microflora have been the subject of numerous studies (15, 16). The halophiles from hypersaline niches can promote the sustainable cultivation of crops in highly saline soils as they are capable of thriving under extreme salt stresses.

In recent years, salt-tolerant plant growth-promoting rhizosphere (PGPR) microorganisms have been used as bioinoculants for the improvement of crop productivity and soil fertility in salt-affected areas. There are several literature reports indicating their positive influence on the growth and yield of various crops under saline conditions (17–20). The PGPRs isolated from saline areas are well adapted to high salt concentrations due to the presence of specialized transporters in their membranes and the synthesis of osmoprotectants for the maintenance of osmotic equilibrium (21). Many PGPRs isolated from the rhizosphere of the plants growing in hypersaline environments have been reported to enhance the growth of plants (22, 23). Although there are several reports on the application of microorganisms isolated from plant rhizosphere of hypersaline ecosystems (22, 23), comprehensive information on the distribution of plant growth-promoting properties across their entire microbial ecosystems is still lacking. Therefore, in the

present study, metagenomic analysis of two hypersaline ecosystems, namely Sambhar Lake and Drang Mine, has been carried out with the aim of (i) studying the adaptational strategies evolved by the microflora of two hypersaline ecosystems to cooperate with prevalent stresses; (ii) investigate the distribution of plant growth promoting attributes for their potential application in saline agricultural lands; and (iii) recovering high-quality MAGs and observe the genetic evidence for various adaptations and plant growth promotion.

## RESULTS

### Metagenomic assembly and binning

Samples for shotgun sequencing were collected from Drang Mine and Sambhar Lake, located in Himachal Pradesh and Rajasthan, respectively (see Fig. S1 at https://figshare.com/s/5e0dbdb113e1249742cb). In our previous study, the physicochemical analysis revealed distinct differences between the samples from Sambhar Lake and Drang Mine. Specifically, the Sambhar Lake samples exhibited a higher temperature of 29.8°C ± 0.7°C, an alkaline pH of 9.1 ± 0, lower levels of $Na^+$ ions at 32985.92 ± 437 ppm, higher concentrations of $Mg^{2+}$ ions at 29.55 ± 0.21 ppm, and elevated $K^+$ ion levels at 117.3 ± 1.83 ppm. In contrast, the Drang Mine samples had a lower temperature of 20.4°C ± 0.56°C, a neutral pH of 7.1 ± 0.14, higher $Na^+$ ion concentrations at 50829.30 ± 781.4 ppm, no detectable $Mg^{2+}$ ions, and lower $K^+$ ion levels at 68.07 ± 3.0 ppm (15). The high-quality sequences from all the samples were co-assembled with the help of assembly tools, namely Megahit, MetaSpades, and IDBA-UD (see Fig. S2 at https://figshare.com/s/5e0dbdb113e1249742cb). The assembly through Megahit gives 29059 contigs, with a total length of 147,360,110 bp and an N50 value of 5844 bp. MetaSpades assembly 26166 contigs, a total of 146,369,712 bp with N50 value of 7136 bp. Meanwhile, IDBA-UD gave 19,844 contigs with a total size of 127,969,138 bp and N50 of 8,793 bp (see Fig. S3 at https://figshare.com/s/5e0dbdb113e1249742cb). Binning of each co-assembled sample was first done by MaxBin, MetaBat, CONCOCT and then optimized by DAStool, resulting in 15 MAGs from Megahit, 26 MAGs from MetaSpades, and 26 MAGs from IDBA-UD (see Fig. S6 at https://figshare.com/s/5e0dbdb113e1249742cb). Collectively, 23.52% and 68.6% of the MAGs were contributed by Drang Mine and Sambhar Lake samples, respectively (see Table S1 at https://figshare.com/s/5e0dbdb113e1249742cb).

In total, 67 MAGs were recovered using three individual assembly tools. The genome size of Megahit ranged from 779,377 bp - 4,099,527 bp with N50 values from 4,182 bp to 145,163 bp, MetaSpades ranged from 586,252 bp to 6,720,040 bp with N50 values from 3,153 bp to 68,824 bp, and IDBA-UD ranged from 457,442 bp to 3,417,808 bp with N50 values from 2,150 bp to 54,825 bp. The completeness and the contamination of the MAGs obtained from Megahit, Metaspades, and IDBA-UD varied from 53.29% to 92.73% and 0.09%–26.04%, 12.95%–96.56% and 0%–8.65%, and 26.74%–97.67% and 0%–26.55%, respectively. MAGs meeting the criteria of completion exceeding 90% and contamination below 5% were categorized as high quality. Those falling within the range of completion greater than or equal to 50% and contamination less than 10%, as well as MAGs with completion less than 50% and contamination less than 10%, were designated as medium-quality and low-quality MAGs, respectively, in accordance with the specified criteria outlined by reference 24. Accordingly, the tool Megahit delivered 3 low-quality, 11 medium-quality, and only 1 high-quality MAGs. MetaSpades provided 3 low-quality, 14 medium-quality, and 9 high-quality MAGs. On the other hand, IDBA-UD delivered 5 low-quality, 12 medium-quality, and 9 high-quality MAGs. The CheckM tool assesses the basic statistics of MAGs and assigns the lineages by utilizing their clade-specific marker genes. Among the MAGs assembled by Megahit, seven were from the bacterial, and eight were from the archaeal lineages. The Metaspades delivered 8 MAGs belonging to bacteria and 18 to Archaea. Meanwhile, IDBA-UD assembled 7 MAGs, which were bacteria, and 19 MAGs, which were Archaea (Fig. 1).

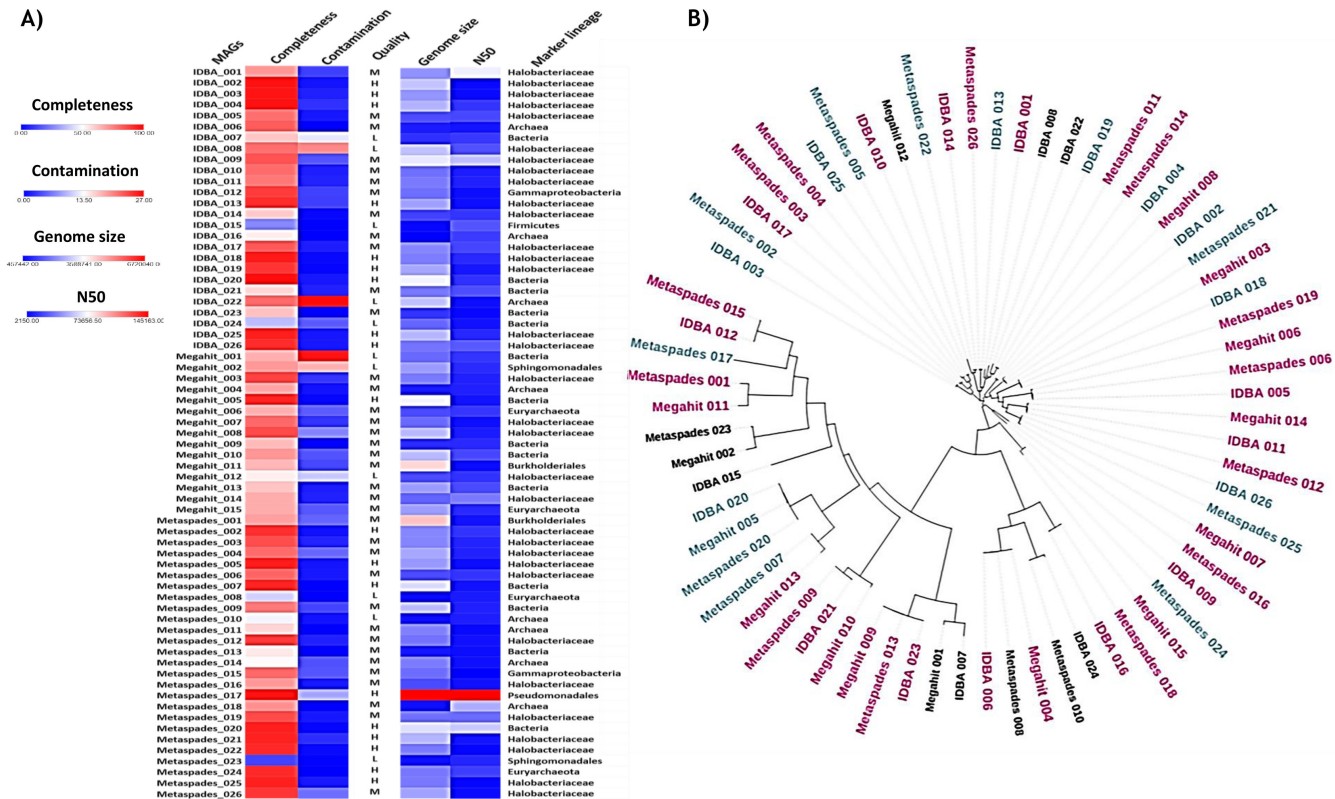

**FIG 1** of the reconstructed MAGs. (A) Heatmap showing the different values of MAGs quality: completeness, contamination, genome size, and N50 values. (B) Phylogenetic visualization of recovered MAGs, where black-colored branches represent low-quality, pink-colored branches represent medium-quality, and blue-colored branches show high-quality MAGs.

## Microbial taxonomic assignment and functional annotation of medium- and high-quality MAGs

The medium- and high-quality groups of the MAGs were aligned against the GTDB database for taxonomic assignment. Out of all the MAGs obtained from three assembly tools after binning, 71.43% and 28.57% of MAGs were assigned to the domains Archaea and Bacteria, respectively. Five of the 12 mid- and high-quality MAGs recovered from the Megahit assembly were classified as genus *Salinarchaeum*. In bacteria, only two MAGs were classified to genera level, one to *Longimonas* and the other to *Massilia*. From assembly carried out by MetaSpades, five MAGs classified as genus *Salinarchaeum*, four MAGs as N*atronomonas,* one as *Halorubrum,* one as *Halovenus*, and one as *Halalkaliarchaeum* belonging to the domain Archaea were recovered. On the other hand, from the domain Bacteria, one *Massilia*, one *Spiribacter,* one *Pseudomonas,* and one *Longimonas* genera were recovered. By contrast, in the IDBA-UD assembly, five MAGs were assigned to genera *Salinarchaeum*, four MAGs to N*atronomonas,* two to *Halorubrum,* one to *Halovenus*, and one to *Halalkaliarchaeum*. Meanwhile, in Bacteria, only two MAGs belonging to the genera *Spiribacter* and *Longimonas* were obtained (Fig. 2). To understand the putative functions of the dominant bacteria in hypersaline ecosystems, the medium- and high-quality MAGs were functionally annotated using the KEGG database (25). The functional profiling revealed the prevalence of salt-tolerant and heavy metal-tolerant strategies in their residential microflora. Approximately 83.33% of the MAGs recovered by Megahit, 91.3% by MetaSpades, and 95.23% by IDBA-UD showed genes related to salt stress. The MAGs IDBA_002, IDBA_013, IDBA_026, Megahit_013, and Metaspades_012 exhibited the presence of all the selected salt-tolerant genes (Fig. 2). The heavy metal stress tolerance-associated genes were found across all the MAGs obtained from Megahit, 95.65% of the MAGs obtained from MetaSpades, and 95.23%

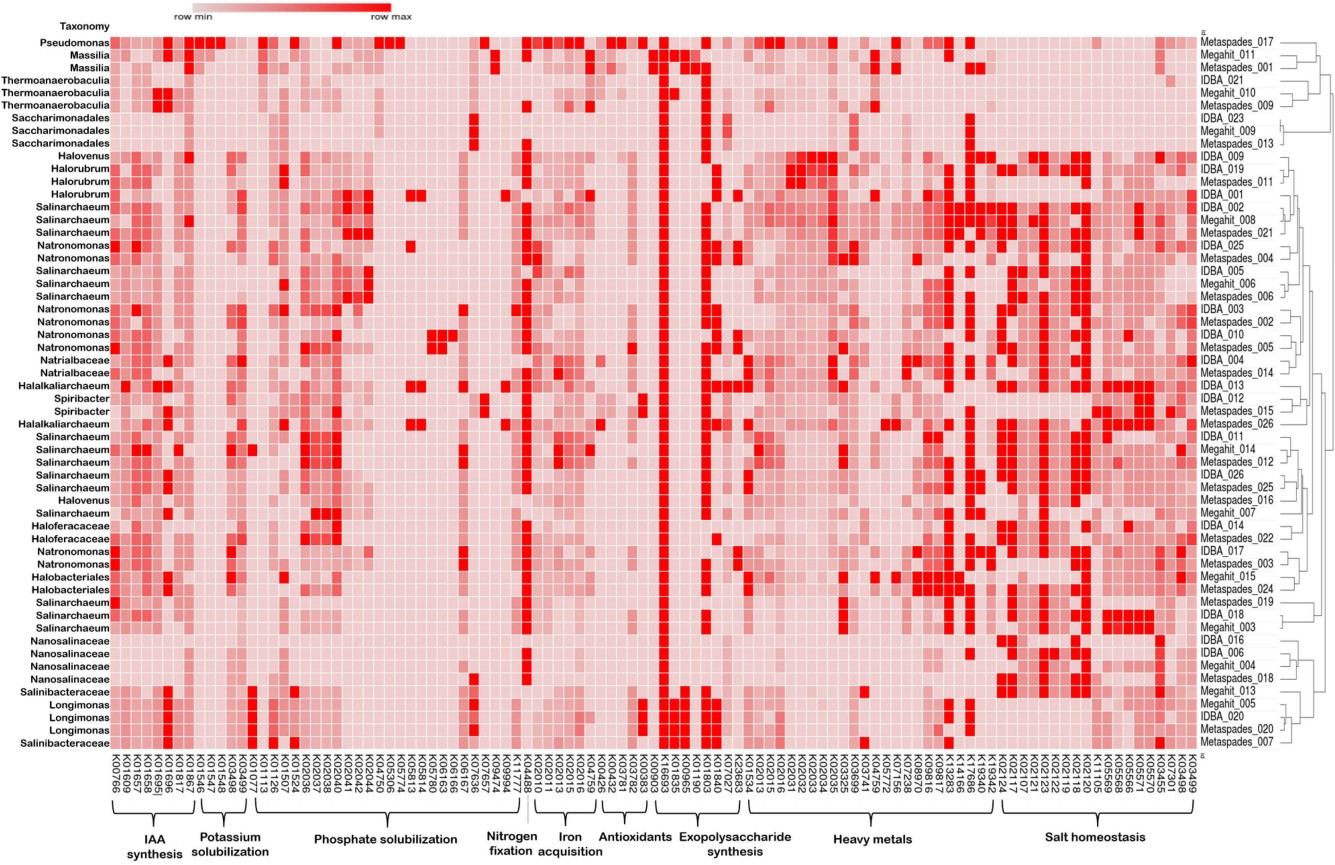

**FIG 2** Taxonomic classification and functional annotation of medium- and high-quality MAGs. The functional analysis revealed genetic evidence for salt tolerance, heavy metal tolerance, and plant growth-promoting potential of hypersaline microflora. The right side of the heatmap displays the hierarchical clustering of the MAGs based on the occurrence of the different genetic attributes.

of the MAGs obtained from IDBA-UD. In addition, some other adaptations, such as exopolysaccharide biosynthesis and antioxidant production, were shown by 96.42% and 64.28% of the total mid- and high-quality MAGs, respectively. The plant growth-promoting (PGP) attributes were observed to be uniformly distributed within the microbiome of hypersaline niches. Specifically, the potential for phosphate solubilization was identified in 100% of the MAGs obtained from both Megahit and MetaSpades, while IDBA-UD revealed this capability in 95.23% of the MAGs.

The iron acquisition was observed in 91.07%, nitrogen fixation in 71.42%, potassium solubilizing in 85.71%, and IAA synthesizing capacity in 98.21% of the total selected MAGs recovered from three different assemblers (see Table S5 at https://figshare.com/s/fe705dbf825508f6d8a3).

## Pangenomic analysis and functional profiling of the selected MAGs)

The MAGs, IDBA_017, Metaspades_003, and Metaspades_015, showing an average nucleotide identity (ANI) of >95%, were then analyzed through pangenome analysis. Upon analysis, IDBA_017 and Metaspades_003 gave the ANI value of 97.62 and 97.6, respectively, to *Natronomonas pharaonis* (Fig. 3A). On the other hand, Metaspades_015 showed an ANI value of 95.01 with *Spiribacter* 2438 (Fig. 3B). The phylogenetic analysis of IDBA_017 and MetaSpades_003 revealed their taxonomic classification as *Natronomonas pharaonis* DSM 2160. In addition, the analysis identified the phylogenetic placement of MetaSpades_015 as *Spiribacter* 2438. The MAGs IDBA_017, Metaspades_003, and Metaspades_015 exhibited a varied number of genes, as shown in Tables S2-S4 (see Tables S2-S4 at https://figshare.com/s/5e0dbdb113e1249742cb). The pangenome

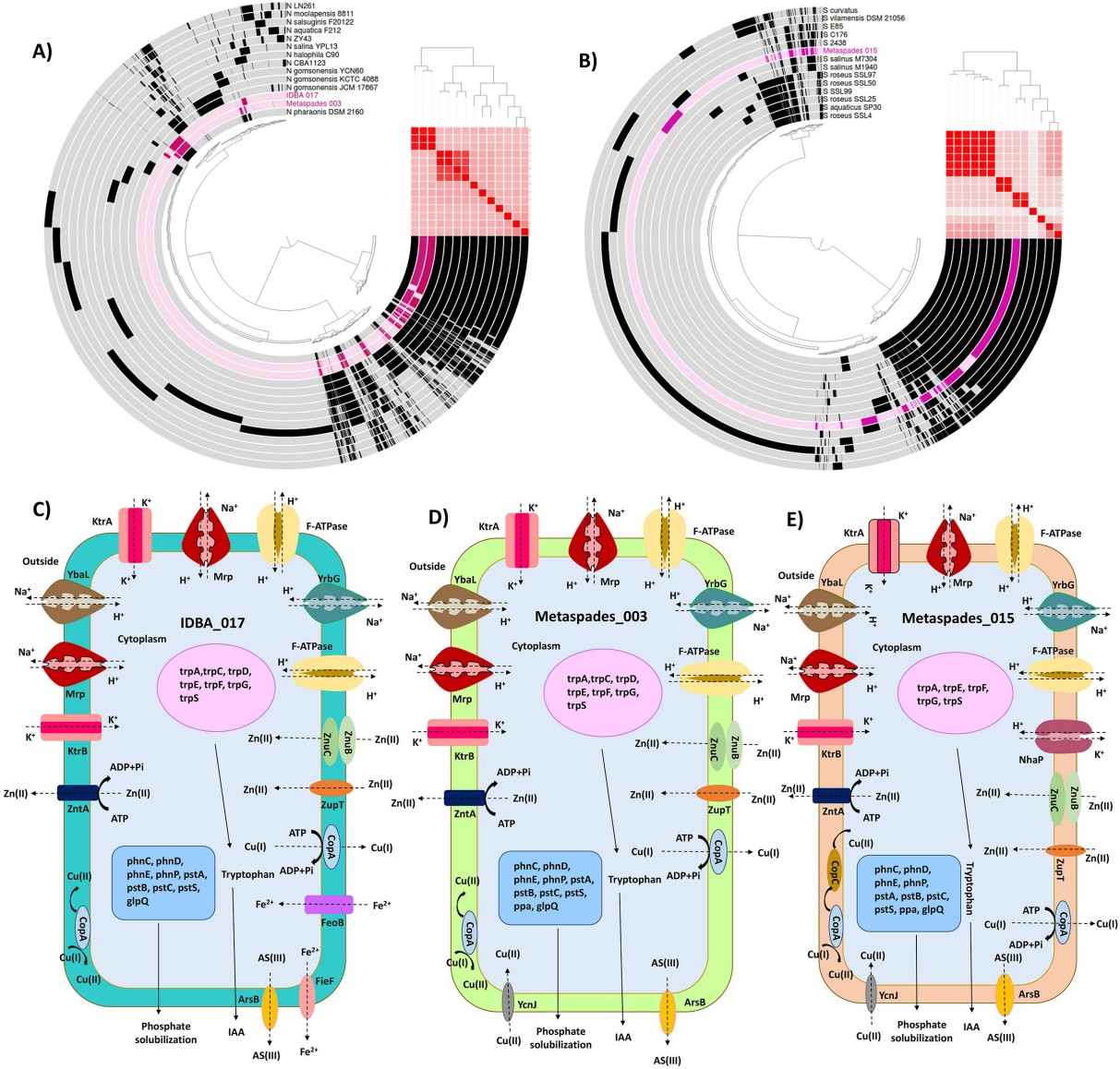

**FIG 3** Pan-genome and average nucleotide identity (ANI) visualization of MAGs with the genome of their respective genera. (A) Comparison of IDBA_017 and Metaspades_003 with the genomes of the genus *Natronomonas*. The MAGs IDBA_017 and Metaspades_003 showed ANI values of 97.62 and 97.6 for *Natronomonas pharaonis*, respectively. (B) Comparison of the MAG Metaspades_015 to the genomes of the genus *Spiribacter*. Metaspades_015 showed the ANI value of 95.01 with *Spiribacter* 2438. (C) Illustration of various salt tolerance, heavy metal stress tolerance, and plant growth-promoting genetic features of IDBA_017, Metaspades_003 (D), and Metaspades_015 (E).

analysis of IDBA_017 and Metaspades_003 with the genome of genus *Natronomonas* indicated the pangenome of 16,025 genes, core genome of 691 genes, Shell genome of 3,670 genes, and cloud genome of 11,664 genes (see Fig. S7 at https://figshare.com/s/5e0dbdb113e1249742cb). A total of 136 and 156 number of unique genes were found in IDBA_017 and Metaspades_003, respectively. Similarly, the pangenome analysis of Metaspades_015 with the genomes present in the genus *Spiribacter* pointed toward a pangenome of 8,711 genes, core genome of 516 genes, Shell genome of 1,625 genes, and cloud genome of 6,570 genes (see Fig. S8 at https://figshare.com/s/5e0dbdb113e1249742cb). Around, 333 unique genes were found in Metaspades_015. Both of the pangenomes were found to be open, which means more genomes can be added.

Upon the functional annotation of the MAGs IDBA_017, Metaspades_003, and Metaspades_015 by the KEGG database, several putative genes related to salt stress tolerance, heavy metal tolerance, and plant growth promotion were observed. The salt-tolerant genes such as *ktr*A, *ktr*B, *yrb*G, *yba*L, *mrp*BDEFG, and F-type ATPases were present in IDBA_017. In Metaspades_003 *ktr*A, *ktr*B, *yrb*G, *yba*L, and *mrp*BDEFG salt-tolerant genes were obtained, whereas, in Metaspades_015, an additional gene *nha*P2 was detected. The rest of the genes were similar to those of Metaspades_003. In IDBA_017, resistance to zinc, iron, and arsenic was found, while in Metaspades_003, resistance to metal zinc, copper, and arsenic was noted. Metaspades_015 exhibited tolerance to zinc, copper, iron, and arsenic. Furthermore, all of these MAGs, IDBA_017, Metaspades_003, and Metaspades_015 possessed potential PGP attributes to various extents (Fig. 3C through E)

## DISCUSSION

Hypersaline environments, characterized by salt concentrations equal to or exceeding the point of salt saturation, harbor diverse microorganisms. Most of their inhabitants are halotolerant or moderate halophiles, with some being extreme halophiles capable of withstanding up to 2.5M NaCl. The microflora residing in hypersaline ecosystems have evolved various adaptive strategies to cope with challenges such as high salt concentrations, osmotic pressure, oligotrophy, and temperature variations, essential for their growth and survival. Notably, these microbes actively contribute to the biogeochemical cycling of different elements, as highlighted in recent studies (15, 16), implying their potential role in promoting plant growth. Given the remarkable ability of halophiles from hypersaline niches to thrive under extreme salt stresses, there is growing interest in exploring their potential for enhancing the sustainable cultivation of crops in highly saline soils. Therefore, the objective of the present study is to investigate the occurrence of various survival mechanisms and the distribution of PGP attributes within the microbial communities of two distinct hypersaline ecosystems, Sambhar Lake and Drang Mine (15), as shown in Fig. S1 (see Fig. S1 at https://figshare.com/s/5e0dbdb113e1249742cb), through genome resolved metagenomic analysis approach.

The genomes from the co-assembled samples of Drang Mine and Sambhar Lake ecosystems were reconstructed via assembly through three advanced metagenomic assembly tools: Megahit, MetaSpades, and IDBA-UD, all of which are de Bruijn graph-based assemblers and iteratively analyze *k*-mer lengths to find the optimal value. In metagenomic studies, the read assembly is a very crucial step for subsequent analysis. The integrity, contiguity, and accuracy of these assemblers vary in relation to many factors, such as the genetic diversity and the sequencing depth (26), so a suitable assembler for a specific data set is necessary to optimize at first. In the current study, the highest assembly length was observed in the case of MetaSpades (146,369,712 bp) as compared to that of Megahit (147,360,110 bp) and IDBA-UD (127,969,138 bp). Although the assembly length was found to be improved in Meta-Spades, the N50 value was highest in IDBA-UD (8793 bp) (see Fig. S3 at https://figshare.com/s/5e0dbdb113e1249742cb). The subsequent genome recovery from each assembly was carried out by frequently used binning tools such as MaxBin2, Meta-Bat2, and CONCOCT. Finally, the bins were optimized by the DAS tool, resulting in the reconstruction of 15 genomes from Megahit, 26 genomes from MetaSpades, and 26 genomes from IDBA-UD assemblies. It was shown that using a variety of binning techniques and combining them can help to rebuild more and higher-quality MAGs from metagenomic data sets (27). DAStool, which selects the best genome from a group of contig-to-bin mappings based on 51 bacterial- and 38 archaeal-specific single-copy marker genes, depends on an iterative dereplication, aggregation, and scoring method. It integrates and optimizes the output from a combination of binning algorithms, which in the present study were MaxBin2, MetaBat2, and CONCOCT (see Fig. S6 at https://figshare.com/s/5e0dbdb113e1249742cb). It generally uses a consensus approach to select non-redundant and high-quality bins (27). The quality of the reconstructed

genome or MAG in terms of completeness and contamination level was assessed by employing CheckM (28). It checks the genome quality by inspecting the marker genes specific to the position of a genome within a reference genome tree. It indicated the recovery of MAGs having the highest quality by MetaSpades followed by IDBA-UD and Megahit assembly according to the standard developed by the Genomic Standards Consortium (24). The obtained number of good quality MAGs simply showed better efficiency of MetaSpades in genome reconstruction over IDBA-UD but high efficiency of both MetaSpades and IDBA-UD over Megahit (Fig. 1). Previously also in a study conducted by Wang and his coworkers, MetaSpades exhibited best performance when compared with other assemblers in comparative metagenomics analysis (26). Another experiment conducted by Van Der Walt and his group also showed a similar kind of observation, so the assembly results obtained in the present study are in accordance with the available literature (29).

The taxonomic classification revealed that the higher numbers of the recovered MAGs comprised of Archaea as compared to Bacteria in the co-assembled Drang Mine and Sambhar Lake samples (Fig. 1). The majority of the MAGs designated to bacterial lineage were classified under phylum Proteobacteria. Zhao et al. also observed Proteobacteria as the dominant phylum in 18 soda-saline lakes in inner Mongolia (30). The *Longimonas*, *Spiribacter*, *Massila,* and *Pseudomonas* were some of the obtained bacterial genera (Fig. 2). The genus *Pseudomonas* possesses a variety of plant growth-enhancing characteristics production of phytohormones (auxins, gibberellins, indole-3-acetic acid), enzymes (aminocyclopropane-1-carboxylate, phenylalanine ammonia-lyase), phosphate & potassium solubilization, and phytopathogen control activities (31). It also plays a crucial role in the metabolism of carbon, nitrogen, sulfur, and arsenic (32, 33). The multifaceted role of *Pseudomonas extends* beyond plant-microbe interactions, encompassing essential functions in various elemental cycles. *Spiribacter*, another genus of interest, demonstrates a specific involvement in the uptake and metabolism of phosphates and phosphonates. Phosphonates are organophosphorus compounds characterized by the presence of a carbon-phosphorus bond and are important in plant growth promotion (34). Furthermore, the genus *Massila* has been identified as possessing lignin-degrading abilities, as demonstrated by Wang and his group (35). The microbial enzymes produced by *Massila* act on lignin, releasing its fractions into the soil. This process not only enhances soil fertility but also provides a nutrient source for neighboring microbes, establishing a collaborative ecosystem within the soil. On the other hand, among Archaea, the majority of the MAGs corresponded to the phylum Euryarcheota. Zhao and his coworkers also found Euryarcheota to be a prevalent group (30). Moreover, the study carried out by Narasingarao and her group also pointed toward the dominance of the Archaeal lineage in hypersaline ecosystems, especially the Euryarcheota phylum (36). The archaea are highly adaptable to harsh conditions and easily attain a stable community structure. The most common archaeal genera observed were *Salinarchaeum*, *Natronomonas*, *Halorubrum*, *Halovenus*, and *Halalkaliarchaeum*. The genus *Salinarchaeum* is well known for its chitinase activity (37). Chitinases play a crucial role in degrading chitin, a polysaccharide present in the exoskeleton of yeast, fungi, and insects. These enzymes contribute to the generation of carbon and nitrogen in ecosystems and find applications in agriculture for controlling various plant pathogens (38). In addition, the genera *Natronomonas and Halorubrum* are involved in the metabolism of nitrogen and sulfur (39, 40). In a study conducted by Garcia-Rolden and her coworkers, they observed a complete dissimilatory nitrate reduction pathway to ammonia in species *N. pharaonic* along with an almost complete assimilatory sulfate reduction pathway converting inorganic sulfate to sulfide in *N. salina* (40). Chen and his group explored the genus *Halorubrum* and identified genes associated with both nitrogen and sulfur metabolism (39). The adequate amount of nitrogen and sulfur has been linked to increased plant growth. Furthermore, the genus *Halorubrum* is known to produce carotenoids and poly(3-Hydroxybutyrate) with diverse biotechnological applications (41, 42). The species *Halovenus aranensis* and *Halovenus carboxidivorans*

have exhibited the production of carotenoids and carbon monoxide oxidizing capacity (43, 44). The genus *Halalkaliarchaeum*, known as sulfur-respiring alkaliphilic haloarchaea, is prevalent in sulfur-rich ecosystems. This genus utilizes elemental sulfur as an electron acceptor to oxidize $CO_2$ through anaerobic carboxydotrophy (45).

Since the microbial diversity in hypersaline ecosystems needs to adjust to different types of stress conditions, it has acquired various types of salt-tolerant mechanisms. The presence of very high salt concentrations in hypersaline niches acts as a driving force for the development of salt-tolerant mechanisms in their residential microbiome. They display a variety of molecular and physiological modifications to inhibit water loss from their cells and maintain osmotic equilibrium with the outside environment. The acquisition of different types of ion-transporting proteins such as $Na^+$ transporters, $Na^+/H^+$ antiporters, $Na^+$/solute symporters, $K^+$ transporters, $K^+/H^+$ antiporters, and $K^+$/solute symporters in their membranes (46). They prevent the unnecessary accumulation of ions inside their cytoplasm, regulating their osmotic pressure and normal functioning. The synthesis of compatible solutes is another crucial strategy developed by halophilic microorganisms. The compatible solutes such as glycine, betaine, ectoine, trehalose, etc., not only regulate the osmotic pressure of the cells but also inhibit the denaturation of macromolecules like proteins, DNA, and RNA at higher salt concentrations (7). Upon investigation, 83.33% of the MAGs recovered by Megahit, 91.3% by MetaSpades, and 95.23% by IDBA-UD assembly possessed genes responsible for salt homeostasis, ensuring their survivability under saline and hypersaline conditions (Fig. 2, see Table S5 at https://figshare.com/s/fe705dbf825508f6d8a3 ). In a previous analysis Sun et al. also noted a variety of salt-tolerant approaches (47). In the present study, the ability of the microbes to tolerate salt stress is accompanied by their heavy metal bioremediation attributes. Approximately 83.33% of the MAGs obtained by Megahit, 86.95% by MetaSpades, and 90.47% by IDBA-UD contained both salt homeostasis and heavy metal tolerance capacity. Some of the earlier studies also reported the coexistence of these properties, improving the environmental fitness and the survivability chances of the microorganisms in highly saline habitats (48).

In addition, the exopolysaccharide and antioxidant biosynthetic potential was also observed in the MAGs obtained from Drang Mine and Sambhar Lake ecosystems (Fig. 2). The exopolysaccharide and antioxidant biosynthesis are also related to the occurrence of stress conditions in hypersaline niches and improve the fitness of their residential microbes. The production of exopolysaccharide (EPS) helps microorganisms to form biofilm and get attached to different types of surfaces. In environments consisting of high concentrations of heavy metals, the extra polymeric substances secreted by the microorganisms adsorb heavy metals and aid in their bioremediation (49, 50). The stressful environmental conditions induce an oxidative response in microorganisms, upregulating the expression of antioxidant genes such as peroxidase, catalase, and superoxide dismutase. The occurrence of different types of ions, minerals, and metals stimulated the hypersaline microflora for active biogeochemical cycling of methane, nitrogen, sulfur cycle, etc., as well (15, 16). The microorganisms in such niches also possess a variety of PGP properties; for instance, in the present study, the microflora displayed iron acquisition, phosphate solubilizing, potassium solubilizing, and IAA synthesizing ability (Fig. 2, see Table S5 at https://figshare.com/s/fe705dbf825508f6d8a3). Iron, phosphate, and potassium are important nutrients playing an important role in plant growth, and their enhanced uptake promotes plant growth and yield. In the literature, some reports related to the prevalence of phosphate solubilizing capacity of hypersaline habitats are also present (51, 52). IAAs serve as an auxin regulating plant growth and development. Several examples of the rhizosphere halophiles exhibiting siderophore production, potassium solubilization, and IAA synthesis were also observed (53, 54). The production of EPS is also a crucial PGP attribute because it facilitates the initial root colonization by microbes and the production of a water-rich polysaccharide layer around roots, which acts as a barrier preventing them from excess ionic salts. This layer also serves as a site for symbiotic

association with other microorganisms and nutrient cycling (55). EPS also aggregates soil particles together, enhances the water absorption capacity of the soil, quorum sensing and maintains the diversity of microorganisms in high salinity conditions (56). The stressed conditions stimulate the generation of reactive oxygen species (ROS) both in microbes and plants. In such conditions, to enhance the tolerance of ROS species in plants, the PGP microorganisms trigger the defense system of the plant, leading to the synthesis of enzymes such as peroxidase, catalase, and superoxide. The IDBA_017 and Metaspades_003 showed >95% ANI value with *Natronomonas pharaonis* upon classification by GTDB toolkit. Their phylogenetic and pangenome analyses with other genomes identified them as *Natronomonas pharaonis* DSM 2160 (Fig. 3A). Similarly, Metaspades_015 showed >95% ANI with *Spiribacter* and was identified as *Spiribacter* 2438 (Fig. 3B). Both constructed pangenomes were open type, and more genomes are simply needed to sequence (see Fig. S8 at https://figshare.com/s/5e0dbdb113e1249742cb). The detailed functional analysis of the MAGs IDBA_017 and Metaspades_003, and Metaspades_015 also showed the presence of a number of adaptations as illustrated (Fig. 3C through E). The genes *ktr*A, *ktr*B, *yba*L, and potassium-chloride symporters were observed for the exchange of $K^+$ and $Cl^-$ as reported previously (57). The *yrb*G or *nha*P and *mrp*BDEFG are involved in the transportation of $Na^+$ ions (58, 59). Besides, several genes related to the resistance against zinc, copper, iron, arsenic, and plant growth enhancement were also observed further supporting earlier findings of the present study.

Overall, it can be said that microorganisms of the hypersaline habitats have developed several different adaptational strategies against salt and heavy metal stresses. The genes providing tolerance to both these stresses co-occur in halophiles. The hypersaline microflora is also rich in genes responsible for plant growth promotion; hence, they can act as a significant bio-stimulating agent. The present study has provided insights into various adaptational strategies and the application of the hypersaline microflora, encouraging their utilization for reclamation and enhanced plant productivity in salinized soils. Furthermore, the utilization of a MAGs-based approach to study environments, such as hypersaline ecosystems, may open new opportunities for exploring and understanding diverse microbial niches and their biotechnological applications.

## MATERIALS AND METHODS

### Sample collection and DNA extraction

The halite fragments were collected from Drang Mine (31.80443° N 76.94636° E) in October 2021. The sediment samples were collected from Sambhar Lake (26.92961° N 75.17642° E) in May 2020. Approximately five samples were collected from each sampling site in sterilized plastic bottles. The samples were transported to the laboratory in an icebox for analysis. The extraction of DNA was carried out according to the protocol mentioned in our previous study (15). Briefly, the halite fragments were crushed with the help of a sterilized mortar and pestle. Each sample was taken in 5 g (halite and sediment) and sieved to remove any larger particles. After that, 15 mL of the extraction buffer (100 mM Tris-HCl, 100 mM sodium phosphate, 1.5 M NaCl, 1% CTAB, and pH 8.0) was added, and the mixture was incubated at 37°C for 2 hours at 120 rpm. The FastDNA spin kit (MP Biomedical, USA) was used to extract DNA according to the manufacturer's instructions. The DNA extracted from each sample's replicates was pooled to constitute a homogenous and representative bacterial diversity of a sampling site (60). DNA quality and quantity estimation was performed using NanoDrop One (Thermofisher Scientific, USA) and Qubit Fluorimeter v.3.0 (Thermofisher Scientific, USA). The integrity of the extracted DNA was checked by Nanodrop and on a 2% agarose gel. The physicochemical properties of the samples, such as temperature, pH, electrical conductivity, sodium, potassium, and magnesium ions, were investigated in our previous research (15). Briefly, the individual samples were combined, and the temperature of

each sample was documented on-site using the MAXTECH Multi thermometer during the sampling process. In addition, the pH and electrical conductivity (EC) of the samples were measured in the laboratory, utilizing the Cyberscan 510 pH meter (Thermo Fischer Scientific, US) and a Century digital conductivity meter. Furthermore, the concentrations of sodium, magnesium, potassium, and calcium ions in the samples were determined using a QTEGRA-ICP-MS instrument (Thermo Fischer Scientific, USA).

## Library preparation and metagenomic sequencing

The paired-end library preparation of the extracted DNA was carried out using the TrueSeq Nano library preparation kit (Illumina, USA) for $2 \times 150$ bp chemistry according to the manufacturer's protocol. Briefly, 250 ng of the DNA was sheared by passing it through an M220 tube (Covaris, USA) to generate a fragment of approximately 350 bp. The 3′ and the 5′ end overhangs of the sheared DNA were subjected to end-repair followed by adaptor ligation, ensuring lower rates of the chimera generation. The size selection of the ligated product was done using AMPure XP beads. The size-selected products were then amplified through polymerase chain reaction (PCR) with the help of index primers. The PCR-enriched libraries were analyzed on 4200 Tape Station (Agilent Technologies, USA) using high sensitivity D1000 Screen tape. After obtaining the appropriate Qubit concentration and the mean peak sizes from the Tape station, the prepared libraries were loaded to Novaseq 6000 for cluster generation and sequencing. The paired-end sequencing allowed the template fragments to sequence in both forward and reverse directions. The samples were then allowed to bind the complementary oligos on the flow cell. The adaptors were designed to allow selective cleavage of the forward strand after re-synthesis of the reverse strand during sequencing. The copied reverse strand was then used to sequence the opposite end of the fragment.

## Quality control and binning of (MAGs)

The files obtained after sequencing were demultiplexed using an in-house script. The raw read files were first checked for their quality by FastQC (61). From FastQC output, quality visualization reads properties like base quality score distribution, sequence quality score distribution, average base content per read, GC distribution in the reads, PCR amplification issue, over-represented sequences, and adapter contamination were accessed. Based on the quality report of FastQC, quality filtering, and trimming were carried out with the help of the FASTX toolkit (62). The quality filtration step allows the retention of only high-quality reads (Phred value ≥30). At the same time, quality trimming was necessary to remove adaptors from the reads. Furthermore, any vector or contamination from other sources, occurring generally while sequencing, was done using FastQ Screen (63). The assembly of the reads was carried out through three different individual tools: MEGAHIT v1.2.9 (64), MetaSpades v3.15.3 (65), and IDBA-UD v1.1.3 (66). The binning of the assembled reads was performed by MaxBin v2 (67), MetaBAT v2 (68), and CONCOCT (69). The bins of each assembly obtained from these tools were then aggregated for subsequent optimization by DAStool (27). The genome quality and completeness were assessed by CheckM (28) with default parameters. The statistics of MAGs and the number of contigs were analyzed using QUAST (70).

## Taxonomic classification and functional annotation of (MAGs)

The phylogenetic classification of the selected medium- and high-quality MAGs was done using the GTDB toolkit (71). It utilizes a set of 120 bacterial gene markers to place MAGs on a specific position of the reference tree constructed using the GTDB database (72) by employing both FastANI (73) and pplacer (74). The prediction of the genes was done by Prokaryotic Genome Annotation Pipeline (PGAP) version 3.1 (75) and Prokka (76). The software Barrnap (77) was used to predict the 16S rRNA gene from MAGs. The annotation of the predicted genes was done by implementing the KEGG database (78). The specific functional gene and their functional IDs were filtered manually for

further analysis. The MAGs with average nucleotide Identity (ANI) >95% were subjected to pangenome analysis to assess the pan, core, and unique genes. The selected MAGs and the genomes of respective *Natronomonas* and *Spiribacter* genera were imported in anvi'o 7.1 (79). The average nucleotide identities (ANIs) among genomes were calculated by "anvicompute-genome-similarity" with PyANI (80). The database of all the genomes was generated using "anvi-gen-genomes-storage," and the pangenome was constructed using "anvi-pan-genome." The visualization of the pan-genome was done with "anvi-display-pan." The functional annotation was carried out using the KEGG database (25) in each genome and obtained through "anvi-summarize."

## ACKNOWLEDGMENTS

K.D. is thankful to CSIR, Govt. of India for the "Research Fellowship" Grant CSIR-NET JRF award no: 31/054(0139)/2019-EMR-I/CSIR-NET JRF JUNE 2017. R.K. acknowledges Science and Engineering Research Board Start-up research grant no. SRG/2019/001071, DST-TDT project no. DST/TDT/WM/2019/43, and CSIR In-House project MLP 0201 for support in conducting research work.

The authors are grateful for the assistance provided by Kishan Kharka, Rahul Kumar for sample collection and Ayush Lepcha for manuscript editing.

This paper represents CSIR-IHBT communication no. 5479.

## AUTHOR AFFILIATIONS

[1]Biotechnology Division, CSIR-Institute of Himalayan Bioresource Technology, Palampur, Himachal Pradesh, India
[2]Academy of Scientific and Innovative Research (AcSIR), Ghaziabad, India

## AUTHOR ORCIDs

Kiran Dindhoria http://orcid.org/0000-0002-1362-0036
Raghawendra Kumar http://orcid.org/0000-0002-7295-7827
Rakshak Kumar http://orcid.org/0000-0002-6982-2454

## AUTHOR CONTRIBUTIONS

Kiran Dindhoria, Conceptualization, Data curation, Formal analysis, Investigation, Methodology, Software, Validation, Writing – original draft | Raghawendra Kumar, Formal analysis, Methodology | Bhavya Bhargava, Visualization, Writing – review and editing | Rakshak Kumar, Conceptualization, Funding acquisition, Supervision, Validation, Writing – review and editing

## DATA AVAILABILITY

The fastq files generated in this analysis were submitted to Sequence Read Archive (SRA), NCBI, to obtain the SRA with accession numbers SRR24060162 and SRR23595080.

## ADDITIONAL FILES

The following material is available online.

Open Peer Review

**PEER REVIEW HISTORY (review-history.pdf).** An accounting of the reviewer comments and feedback.

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
