## [Reviewer comments · mSystems]

Metagenomic assembled genomes indicated the potential application of hypersaline microbiome for plant growth promotion and stress alleviation in salinized soils

Kiran Dindhoria, Raghawendra Kumar, Bhavya Bhargava, and Rakshak Kumar

Corresponding Author(s): Rakshak Kumar, Institute of Himalayan Bioresource Technology CSIR

Review Timeline:

Submission Date:	October 3, 2023
Editorial Decision:	November 24, 2023
Revision Received:	January 5, 2024
Accepted:	January 19, 2024

Editor: Rup Lal

Reviewer(s): Disclosure of reviewer identity is with reference to reviewer comments included in decision letter(s). The following individuals involved in review of your submission have agreed to reveal their identity: Shashi Kant Bhatia (Reviewer #1); Mukesh Kumar Awasthi (Reviewer #2)

Transaction Report:

DOI: <https://doi.org/10.1128/msystems.01050-23>

Re: mSystems01050-23 (Metagenomic assembled genomes indicated the potential application of hypersaline microbiome for plant growth promotion and stress alleviation in salinized soils)

Dear Dr. Rakshak Kumar:

Revision Guidelines

Sincerely,
Rup Lal
Editor
mSystems

Reviewer #1 (Comments for the Author):

This study provides a detailed and comprehensive analysis of metagenome-assembled genomes (MAGs) from hypersaline ecosystems, shedding light on the microbial communities and their functional potential in these environments. The research appears to be well-structured, presenting results, methods, and discussion in a logical sequence. The high-quality

sequencing data, bioinformatic analysis, and functional annotation of MAGs offer valuable insights into the adaptation strategies and plant growth potential of microorganisms in hypersaline ecosystems. However, there are some suggestions from my side to improve this manuscript further:

1. In the introduction, consider providing an overview of the importance and ecological relevance of hypersaline ecosystems rather than saline soils. Why are hypersaline ecosystems significant, and what are the key challenges microorganisms face in such environments?
2. The taxonomic classification indicated that the majority of the recovered MAGs were affiliated with Archaea. Authors may include the ecological roles played by these Archaea in hypersaline ecosystems, and how do they interact with other microorganisms in these environments under discussion section?
3. Authors may also calculate bin coverage in order to get better idea of microbial prevalence.
4. In the discussion section, consider providing more context and references to support your findings. This can help readers better understand the significance of your results in the broader context of hypersaline ecosystems and microbial ecology.
5. The authors should pay attention to language and grammatical errors in the manuscript to enhance its readability and clarity.

Reviewer #2 (Comments for the Author):

This study provides a comprehensive analysis of metagenome-assembled genomes (MAGs) from hypersaline ecosystems, yielding valuable insights into microbial community composition and functional capabilities. The study elucidates the adaptive strategies of microorganisms in hypersaline environments, with a specific focus on their potential to support plant growth.

However, there are some comments to enhance the manuscript from my side:

The introduction should be more focused with proper citations of other related studies from hypersaline ecosystems.

The metagenome assembly in present study has been carried out independently using three different assembly tools. The authors should clearly state the parameters for bin quality in discussion. The information regarding the total number of genes, ORFs, and hypothetical genes can also be added.

The details of the genes observed should be added in the form of the table either in main text or in supplementary.

Authors can discuss the functional role of the observed taxa in context of the hypersaline ecosystems.

Authors should focus on improving the language of the manuscript.

This study provides a comprehensive analysis of metagenome-assembled genomes (MAGs) from hypersaline ecosystems, yielding valuable insights into microbial community composition and functional capabilities. The study elucidates the adaptive strategies of microorganisms in hypersaline environments, with a specific focus on their potential to support plant growth. However, there are some comments to enhance the manuscript from my side:

The introduction should be more focused with proper citations of other related studies from hypersaline ecosystems.

The metagenome assembly in present study has been carried out independently using three different assembly tools. The authors should clearly state the parameters for bin quality in discussion. The information regarding the total number of genes, ORFs, and hypothetical genes can also be added.

The details of the genes observed should be added in the form of the table either in main text or in supplementary.

Authors can discuss the functional role of the observed taxa in context of the hypersaline ecosystems.

Authors should focus on improving the language of the manuscript.

Reviewer #1 (Comments for the Author):

This study provides a detailed and comprehensive analysis of metagenome-assembled genomes (MAGs) from hypersaline ecosystems, shedding light on the microbial communities and their functional potential in these environments. The research appears to be well-structured, presenting results, methods, and discussion in a logical sequence. The high-quality sequencing data, bioinformatic analysis, and functional annotation of MAGs offer valuable insights into the adaptation strategies and plant growth potential of microorganisms in hypersaline ecosystems. However, there are some suggestions from my side to improve this manuscript further:

Comment 1: In the introduction, consider providing an overview of the importance and ecological relevance of hypersaline ecosystems rather than saline soils. Why are hypersaline ecosystems significant, and what are the key challenges microorganisms face in such environments?

Response 1: Thank you for bringing this to our notice. Upon your suggestion, the overview of the importance and ecological relevance of hypersaline ecosystems and their key challenges have been incorporated in the introduction section of the manuscript as follows:

“Hypersaline environments are..... plant growth promotion.”

Comment 2: The taxonomic classification indicated that the majority of the recovered MAGs were affiliated with Archaea. Authors may include the ecological roles played by these Archaea in hypersaline ecosystems, and how do they interact with other microorganisms in these environments under discussion section?

Response 2: Thanks for your suggestion. The ecological roles of the MAGs affiliated with Archaea and their interactions with other microbes are now incorporated in a paragraph under discussion section as follows:

“On the other hand, among Archaea anaerobic carboxydofrophy.”

Comment 3: Authors may also calculate bin coverage in order to get better idea of microbial prevalence.

Response 3: Thanks for your valuable suggestion. We have now calculated the coverage of each bin in both samples and incorporated the table in the supplementary information as Table S1. We have also cited the table at relevant positions in the manuscript.

Comment 4: In the discussion section, consider providing more context and references to support your findings. This can help readers better understand the significance of your results in the broader context of hypersaline ecosystems and microbial ecology.

Response 4: Thank you for pointing this out. Upon your suggestion, we have incorporated new references to support our findings as follows:

“In a study conducted by Wang et al., in 2020..... with the available literature.”

“Zhao et al., in 2020, also foundthe Euryarcheota phylum”

“In a previous analysis by Sun et al., in 2022.....salt-tolerant approaches.”

“Some of the earlier highly saline habitats (Rathore et al., 2021).”

Comment 5: The authors should pay attention to language and grammatical errors in the manuscript to enhance its readability and clarity.

Response 5: Thank you for the valuable suggestion. The Premium version of the Grammarly software is implemented to make corrections to the revised manuscript.

Reviewer #2 (Comments for the Author):

This study provides a comprehensive analysis of metagenome-assembled genomes (MAGs) from hypersaline ecosystems, yielding valuable insights into microbial community composition and functional capabilities. The study elucidates the adaptive strategies of microorganisms in hypersaline environments, with a specific focus on their potential to support plant growth. However, there are some comments to enhance the manuscript from my side:

Comment 1: The introduction should be more focused with proper citations of other related studies from hypersaline ecosystems.

Response 1: Thank you for bringing this to our notice. We have now included the citations of recent studies from hypersaline ecosystems in a paragraph under introduction section as shown below:

“There are several reports..... numerous studies (Dindhoria et al., 2023a; Liu et al., 2023).”

Comment 2: The metagenome assembly in present study has been carried out independently using three different assembly tools. The authors should clearly state the parameters for bin quality in discussion. The information regarding the total number of genes, ORFs, and hypothetical genes can also be added.

Response 2: The parameters used for assigning the bin quality are now incorporated in the results and discussion sections. The information regarding genes such as CDS, hypothetical genes and tRNAs of the selected MAGs is now added in the supplementary sheet as Table S2- S4. The information regarding parameters used for assigning the bin quality is incorporated as follows:

Result section:

“MAGs meeting the criteria of outlined by Bowers et al., in 2017.”

Discussion section:

“It indicated the recovery of MAGsstandard developed by the Genomic Standards Consortium (Bowers et al., 2017).”

Comment 3: The details of the genes observed should be added in the form of the table either in main text or in supplementary.

Response 3: Thank you for pointing this out. We have now added a table containing all the observed genes in all MAGs in the supplementary information as Table S5.

Comment 4: Authors can discuss the functional role of the observed taxa in context of the hypersaline ecosystems.

Response 4: Thanks for your suggestion. The functional roles of the observed taxa have now been added to the discussion section as follows:

“The genus *Pseudomonas*.....anaerobic carboxydofrophy.”

Comment 5: Authors should focus on improving the language of the manuscript.

Response 5: Thank you for the valuable suggestion. The Premium version of the Grammarly software is implemented to make corrections to the revised manuscript.

Re: mSystems01050-23R1 (Metagenomic assembled genomes indicated the potential application of hypersaline microbiome for plant growth promotion and stress alleviation in salinized soils)

Dear Dr. Rakshak Kumar:

Your manuscript has been accepted, and I am forwarding it to the ASM production staff for publication. Your paper will first be checked to make sure all elements meet the technical requirements. ASM staff will contact you if anything needs to be revised before copyediting and production can begin. Otherwise, you will be notified when your proofs are ready to be viewed.

Featured Image Submissions: If you would like to submit a potential Featured Image, please email a file and a short legend to msystems@asmusa.org. Please note that we can only consider images that (i) the authors created or own and (ii) have not been previously published. By submitting, you agree that the image can be used under the same terms as the published article. Image File requirements: TIF/EPS, 7.5 inches wide by 8.25 inches tall (at least 2,250 pixels wide by 2,475 pixels tall), minimum 300 dpi resolution (600 dpi preferred), RGB, and no figure elements, e.g., arrows or panel labels. The legend should be a short description of the image, 1-2 sentences recommended.

We recognize that the video files can become quite large, so to avoid quality loss ASM suggests sending the video file via <https://www.wetransfer.com/>. When you have a final version of the video and the still ready to share, please send it to mSystems staff at msystems@asmusa.org.

Sincerely,
Rup Lal

Editor
mSystems

Reviewer #1 (Comments for the Author):

The authors have revised the manuscript well according to all the reviewer's comments.

Reviewer #2 (Comments for the Author):

Please accept in present form because authors have properly addressed all the comments.